# Network analyses identify liver-specific targets for treating liver diseases

Sunjae Lee[1,†] , Cheng Zhang[1,†] , Zhengtao Liu[1,†], Martina Klevstig[2], Bani Mukhopadhyay[3], Mattias Bergentall[2], Resat Cinar[3] , Marcus Ståhlman[2], Natasha Sikanic[1], Joshua K Park[3], Sumit Deshmukh[1], Azadeh M Harzandi[1], Tim Kuijpers[1], Morten Grøtli[4], Simon J Elsässer[5], Brian D Piening[6], Michael Snyder[6], Ulf Smith[2], Jens Nielsen[1,7] , Fredrik Bäckhed[2], George Kunos[3], Mathias Uhlen[1] , Jan Boren[2,*] & Adil Mardinoglu[1,7,**]

## Abstract

We performed integrative network analyses to identify targets that can be used for effectively treating liver diseases with minimal side effects. We first generated co-expression networks (CNs) for 46 human tissues and liver cancer to explore the functional relationships between genes and examined the overlap between functional and physical interactions. Since increased *de novo* lipogenesis is a characteristic of nonalcoholic fatty liver disease (NAFLD) and hepatocellular carcinoma (HCC), we investigated the liver-specific genes co-expressed with fatty acid synthase (FASN). CN analyses predicted that inhibition of these liver-specific genes decreases FASN expression. Experiments in human cancer cell lines, mouse liver samples, and primary human hepatocytes validated our predictions by demonstrating functional relationships between these liver genes, and showing that their inhibition decreases cell growth and liver fat content. In conclusion, we identified liver-specific genes linked to NAFLD pathogenesis, such as pyruvate kinase liver and red blood cell (PKLR), or to HCC pathogenesis, such as PKLR, patatin-like phospholipase domain containing 3 (PNPLA3), and proprotein convertase subtilisin/kexin type 9 (PCSK9), all of which are potential targets for drug development.

**Keywords** co-expression; co-regulation; HCC; metabolism; NAFLD
**Subject Categories** Genome-Scale & Integrative Biology; Molecular Biology of Disease; Network Biology
**Mol Syst Biol. (2017) 13: 938**

## Introduction

Nonalcoholic fatty liver disease (NAFLD) is characterized by the accumulation of excess fat in the liver and is associated with obesity, insulin resistance (IR), and type 2 diabetes (T2D). NAFLD includes a spectrum of diseases ranging from simple steatosis to nonalcoholic steatohepatitis (NASH) and plays a major role in the progression of cirrhosis and hepatocellular carcinoma (HCC), a cancer with one of the highest mortality rates worldwide (Kew, 2010). Although NAFLD is the most common cause of chronic liver disease in developed countries, and its worldwide prevalence continues to increase along with the growing obesity epidemic, there is no approved pharmacological treatment for NAFLD. NAFLD is projected to become the most common indication leading to liver transplantation in the United States by 2030 (Shaker *et al*, 2014). The incidence of HCC has also increased significantly in the United States over the past few decades, in parallel with the epidemic of NAFLD (Petrick *et al*, 2016). Hence, there is an urgent need to develop new strategies for preventing and treating such chronic hepatic diseases.

Biological networks can be used to uncover complex systems-level properties. Systems biology combining experimental and computational biology to decipher the complexity of biological systems can be used for the development of effective treatment strategies for NAFLD, HCC, and other complex diseases (Mardinoglu & Nielsen, 2015; Yizhak *et al*, 2015; Mardinoglu *et al*, 2017b; Nielsen, 2017). To date, several metabolic processes that are altered in NAFLD (Mardinoglu *et al*, 2014a, 2017a; Hyötyläinen *et al*, 2016) and HCC (Agren *et al*, 2012, 2014; Björnson *et al*, 2015; Elsemman *et al*, 2016) have been revealed through the use of genome-scale

1   Science for Life Laboratory, KTH – Royal Institute of Technology, Stockholm, Sweden
2   Department of Molecular and Clinical Medicine, University of Gothenburg and Sahlgrenska University Hospital, Gothenburg, Sweden
3   Laboratory of Physiologic Studies, National Institute on Alcohol Abuse and Alcoholism, National Institutes of Health, Bethesda, MD, USA
4   Department of Chemistry and Molecular Biology, University of Gothenburg, Gothenburg, Sweden
5   Department of Medical Biochemistry and Biophysics, Karolinska Institutet, Stockholm, Sweden
6   Department of Genetics, Stanford University, Stanford, CA, USA
7   Department of Biology and Biological Engineering, Chalmers University of Technology, Gothenburg, Sweden
    *Corresponding author. Tel: +46 31 342 2949; Fax: +46 31 823 762; E-mail: jan.boren@wlab.gu.se
    **Corresponding author. Tel: +46 31 772 3140; Fax: +46 31 772 3801; E-mail: adilm@scilifelab.se
    †These authors contributed equally to this work

metabolic models (GEMs), a popular tool in systems biology. GEMs are reconstructed on the basis of detailed biochemical information and have been widely used to determine the underlying molecular mechanisms of metabolism-related disorders (Mardinoglu & Nielsen, 2012, 2015, 2016; Mardinoglu *et al*, 2013b, 2015; Yizhak *et al*, 2013, 2014a,b; Bordbar *et al*, 2014; Björnson *et al*, 2015; O'Brien *et al*, 2015; Shoaie *et al*, 2015; Zhang *et al*, 2015; Uhlen *et al*, 2016).

Recently, we have integrated GEMs for hepatocytes (*iHepatocytes2322*; Mardinoglu *et al*, 2014a), myocytes (*iMyocytes2419*; Varemo *et al*, 2015), and adipocytes (*iAdipocytes1850*; Mardinoglu *et al*, 2013a, 2014b) with transcriptional regulatory networks (TRNs) and protein–protein interaction networks (PPINs) to generate tissue-specific integrated networks (INs) for liver, muscle, and adipose tissues (Lee *et al*, 2016). The INs allowed us to comprehensively explore the tissue biological processes altered in the liver and adipose tissues of obese subjects, thus accounting for the effects of transcriptional regulators and their interacting proteins and enzymes. Although INs provide physical interactions between pairs or groups of enzymes, transcription factors (TFs), and other proteins, these physical interactions may not necessarily have close functional connections.

Co-expression connections are enriched for functionally related genes, and co-expression networks (CNs) allow the simultaneous investigation of multiple gene co-expression patterns across a wide range of clinical and environmental conditions. In this study, we constructed CNs for major human tissues including liver, muscle, and adipose tissues and studied the overlap between functional connections and physical interactions defined by CNs and INs, respectively. We also constructed CNs for HCC to investigate the functional relationship between genes. Finally, we used these comprehensive biological networks to explore the altered biological processes in NAFLD and HCC and identified liver-specific gene targets that may be used in the development of effective treatment strategies for NAFLD and HCC with likely minimal negative side effects.

# Results

### Generation of CNs for human tissues

A common observation in gene expression analysis performed for different clinical conditions is that many genes known to be functionally related often show similar expression patterns, thus potentially indicating shared biological functions under common regulatory control. Thus, identifying co-expression patterns instead of only differential expression patterns may be informative for understanding altered biological functions. To identify genes with similar gene expression profiles, we retrieved RNA-seq data comprising 51 tissues, including liver, muscle, and subcutaneous and omental adipose, along with other major human tissue samples (Dataset EV1), from the Genotype-Tissue Expression (GTEx) database (GTEx, 2013). To measure the tendency of gene expression correlation, we calculated the Pearson correlation coefficients ($r$) between all gene pairs in 46 human tissues (Dataset EV1) with more than 50 samples, and we ranked all genes according to the calculated $r$. We used the top 1% correlation value of each tissue as a cutoff indicating that two genes were co-expressed (average

$r = 0.576$ for 46 tissues) and therefore had the same number of co-expression links for all tissues, thus yielding 1,498,790 links, and combined them to construct the tissue-specific CNs. The resulting CNs were as follows: Liver tissue contained 11,580 co-expressed genes, muscle tissue contained 10,728 co-expressed genes, and subcutaneous and omental adipose tissue contained 12,120 and 11,117 co-expressed genes (Dataset EV1).

Given the connectivity of tissue-specific co-expression links, we found groups of highly co-expressed genes, termed co-expression clusters, by using the random walk community detection algorithm (Pons & Latapy, 2005; Fig 1A). Among these clusters, we selected the most highly co-expressed key clusters on the basis of their clustering coefficients (average clustering coefficient = 0.522). We found that genes associated with key co-expression clusters in tissues significantly overlapped with tissue-specific genes presented in the Human Protein Atlas (HPA; Uhlen *et al*, 2015), when global proteomics and transcriptomics data were available for more than 30 human tissues. Our analysis indicated that 75.6% of 41 HPA-matched tissues had key co-expression clusters significantly enriched in tissue-specific genes (hypergeometric test $P < 0.01$), thus suggesting the tissue-specific roles of the genes associated with the key clusters (Dataset EV2).

To investigate tissue-specific functions of the genes associated with the key co-expression clusters, we performed gene ontology (GO) enrichment analysis by using the GO biological processes (BP) terms in MSigDB (Subramanian *et al*, 2005; hypergeometric test $P < 1.0 \times 10^{-4}$; Fig EV1). For example, in liver tissue, we found that genes associated with key co-expression clusters were enriched in terms comprising the immune response, hemopoiesis, and fatty acid metabolic processes (Fig 1B), whereas in testis tissue, co-expression clusters were enriched in cell cycle, metabolism, and reproduction (Fig EV1C and Dataset EV3). Likewise, we investigated the enrichment of the genes associated with key co-expression clusters in other metabolically active tissues, including skeletal muscle and subcutaneous and omental adipose tissues, compared with the significantly enriched GO BP terms in liver tissue, and found that the GO BP term "hemopoiesis" was enriched only in liver tissue, whereas "muscle development" was enriched only in skeletal muscle (Fig EV1A). In both subcutaneous adipose tissue and liver tissue, genes associated with the key clusters were significantly enriched in fatty acid metabolic processes (Fig EV1B). Hence, our analysis indicated that genes involved in fatty acid metabolism are significantly co-expressed in liver and adipose tissues.

### Increased co-regulation results in increased co-expression

We have previously presented INs for metabolic tissues, including liver, muscle, and adipose tissues, and have identified the physical links between TFs and target proteins and enzymes (Lee *et al*, 2016). Here, we investigated the overlap between the INs and CNs and analyzed the potential deregulation of metabolism in metabolically active human tissues by using the topology on the basis of the regulatory interactions and protein–protein interactions of INs by comparing the mean co-expression of actual networks with that of networks randomly permuted by 1,000 repetitions (Fig 1C). We found that most of the actual tissue regulatory networks (RNs; Fig 1D) and PPINs (Fig 1E) showed higher co-expression than randomly permutated networks in liver and other metabolic tissues

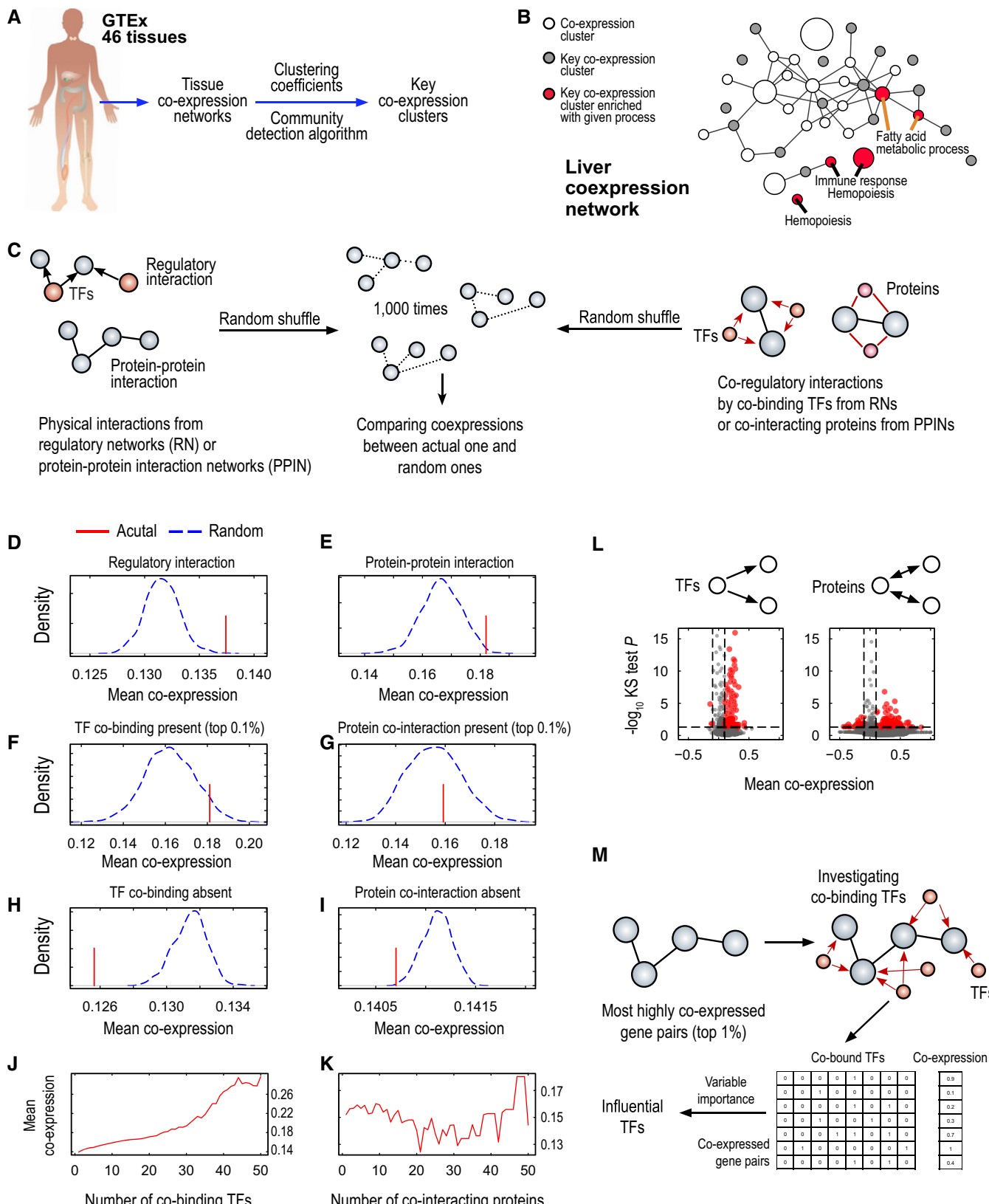

**Figure 1.**

**Figure 1.    Characteristics of tissue-specific co-expression networks (CNs).**

A    We generated co-expression networks for 46 tissues that had more than 50 samples from GTEx RNA-seq data. In each tissue, we found groups of highly co-expressed genes, called co-expression clusters, by using a community detection algorithm. Among these groups, we selected key co-expression clusters on the basis of their clustering coefficients.

B    For example, in liver tissue, we found genes from key co-expression clusters significantly enriched in biological processes required for liver function, such as fatty acid metabolism, hemopoiesis, and immune response. All nodes in the network stand for co-expression clusters, and edges are connected when genes belonging to those clusters were highly connected more frequently than at random. Node sizes are proportional to the number of genes involved in the respective clusters.

C–K  In liver tissue, we investigated co-expression of physical interactions from the liver regulatory network (RN) or liver protein–protein interaction network (PPIN) (red lines). We also generated randomly permutated gene pairs from those physical interactions by 1,000× (blue dashed lines) and compared them with actual gene pairs from RN or PPIN (D and E, respectively). We identified co-regulatory interactions from RN or PPIN (F and G, respectively), which indicate gene pairs co-bound by the same TFs or co-interacting with the same proteins, and compared them with randomly permutated gene pairs; here, we examined only the co-regulated gene pairs with the highest numbers of co-bound TFs or co-interacting proteins (top 0.1%). In addition, we identified gene pairs that had no co-regulation by TF binding or protein interactions (H and I, respectively) and compared them with randomly permutated gene pairs. We found that gene pairs from actual physical interactions (D and E) or co-regulatory interactions (F and G) had higher co-expression than at random; however, gene pairs with no co-regulations (H and I) had lower co-expression than at random. We examined the co-expression profiles of co-regulated gene pairs according to their co-bound TFs (J) or co-interacting proteins (K). Here, we found that only co-regulations from RN were associated with increased co-expression, whereas co-regulations from PPIN were not, thus suggesting that RN has more specificity for increasing co-expression.

L    We examined which TFs or proteins were highly co-expressed with their bound target genes or interacting proteins by comparing their gene pairs with the overall expression by Kolmogorov–Smirnov two-sided test ($P < 0.05$) and the absolute value of mean co-expression ($|r| > 0.1$).

M    To find the most influential TFs in co-expression, we established a feature matrix between highly co-expressed gene pairs (top 1%) and their co-bound TFs and fitted it to a Random Forest model; we considered co-bound TFs as predictor variables and co-expression values as response variables. From this model, we calculated variable importance scores of TFs, and on the basis of these scores, we identified the most influential TFs in co-expression in liver tissue. The top 1% most influential TFs included YY1, CTCF, RAD21, SREBF1, and SREBF2 and were enriched in liver development, interphase of the mitotic cell cycle, and response to lipids.

(Figs EV2A and B, and EV3A and B). We also observed that regulatory interactions built from *in vitro* differentiated adipocytes (Lee *et al*, 2016) had less specificity than co-expression calculated by using subcutaneous adipose tissue RNA-seq data demonstrating higher tissue specificity (Fig EV2B).

Target proteins regulated by the same TFs or interacting with the same proteins may have similar gene expression patterns (Zhang *et al*, 2016). Hence, we examined the co-expression of co-regulated gene pairs in RNs and found that their mean co-expression was higher in actual RNs than in randomly permutated RNs in the liver (Fig 1F) and other metabolic tissues (Fig EV2C and D). Moreover, we examined the co-expression of gene pairs that were not regulated by the same TFs and found that their mean co-expression was lower in actual liver RNs than in randomly permutated RNs in liver tissue (Fig 1H) and other metabolic tissues (Fig EV2E and F). Similarly, we examined the co-expression of co-interacting gene pairs for the same proteins in PPINs and found relatively lower mean co-expression in actual liver PPINs than in randomly permutated PPINs in liver tissue (Fig 1G) and other metabolic tissues (Fig EV3C and D), as compared with the mean co-expression of the actual RNs. We also found that mean co-expression in PPINs was lower in actual PPINs than in randomly permutated PPINs in liver tissue (Fig 1I) and other metabolic tissues (Fig EV3E and F) when we compared proteins that did not interact with the same proteins. Our analysis indicated that physical interactions defined by the RNs and PPINs can be used to explain protein co-expression. Moreover, we observed that the two target proteins regulated by the same TFs may have similar expression patterns, whereas two target proteins interacting with the same protein may have less similar expression patterns.

Target proteins may be regulated by more than one TF in RNs, thus potentially affecting their expression patterns. In this context, we determined the co-expression of two target proteins regulated by the same TFs and found that an increased number of co-bound TFs were likely to be associated with increased mean co-expression levels in liver (Fig 1J), skeletal muscle, and adipose tissues (Fig EV2G and H). We also repeated a similar analysis in PPINs on the basis of the two target proteins that co-interacted with the same proteins and found that the increased number of co-interacting proteins was not directly proportional to increased mean co-expression although the mean co-expression levels were high in the liver (Fig 1K) and other tissues analyzed (Fig EV3G and H). Our analysis indicates that protein interactions provide evidence for increased co-expression by RNs compared with PPINs in metabolic tissues.

We also identified the TFs and proteins that were highly co-expressed with their target genes or proteins in RNs and PPINs, respectively ($|r| > 0.1$ and Kolmogorov–Smirnov (KS) two-sided test, $P < 0.05$) (Fig 1L, Datasets EV4 and EV5). We investigated the biological functions associated with these TFs and proteins using DAVID (Huang *et al*, 2009) and found that the TFs in liver RN were enriched in the regulation of transcription and cell differentiation (false discovery rate [FDR] < 0.01) and that the proteins in liver PPIN were enriched in RNA splicing (FDR < 0.01), which is involved in post-transcriptional regulation. To identify the most influential TFs in the co-expression of gene pairs, we used the most highly co-expressed gene pairs (top 1%), established the co-binding matrix of bound TFs on the basis of RNs, and calculated the variable importance score by using the Random Forest model in the liver (Fig 1M and Dataset EV6). Among the top 1% most influential TFs (76 TFs), for example, YY1, CTCF, RAD21, SREBF1, and SREBF2, we found TFs enriched in liver development, interphase of the mitotic cell cycle, and response to lipids (FDR < 0.001), thus suggesting that many influential TFs are involved in liver-specific functions.

**Highly co-expressed metabolic pathways in the liver**

On the basis of physical and functional links provided by the networks, we examined which metabolic pathways are regulated

**Figure 2.  Highly co-expressed metabolic pathways in liver tissue with their co-regulating TFs.**

A    Among human metabolic reactions with known enzymes (HMR2), we calculated the co-expression of respective enzymes in liver tissue and established a co-expression matrix of those metabolic reactions for liver tissue. Performing hierarchical clustering on the matrix, we found 100 reaction clusters of highly co-expressed enzymes in liver tissue. We compared the mean co-expression profiles of given reaction clusters in liver tissue with the co-expression profiles of clusters in other tissues, such as skeletal muscle and adipose subcutaneous tissues. On the basis of their differential co-expression levels, we identified liver-specific reaction clusters (B) and their co-regulating TFs (C and D).

B    Among the 100 reaction clusters for liver tissue, we selected ten reaction clusters with the most differential co-expression between liver tissue and skeletal muscle tissue (left) or between liver tissue and adipose subcutaneous tissue (right), regarding them as liver-specific reaction clusters, colored red and blue, respectively (purple for those in both cases).

C    We examined co-regulating TFs of liver-specific reaction clusters found in (B). Using the hypergeometric test, we identified co-regulating TFs significantly enriched in given reaction clusters ($P < 0.05$). Here, we found that reaction cluster 14 was enriched in binding of metabolic nuclear receptors, such as PXR, FXR, and RXR, whereas reaction cluster 9 was enriched in binding of SREBF2, a regulator of lipid homeostasis.

D    We found additional evidence of strong regulation in some liver-specific reaction clusters, on the basis of variable TF importance scores. We compared the variable importance scores (Dataset EV6) of enriched TFs in given reaction clusters with the overall score by using Kolmogorov–Smirnov tests and selected significant clusters ($P < 0.25$) as highly regulated reaction clusters, including reaction clusters 9, 14, 40, and 58.

E, F    We identified HCC-deregulated reaction clusters by comparing co-expression levels between liver tissue and HCC tumor tissue similarly to (B). Yellow-colored are HCC-deregulated liver reaction clusters; green-colored are liver-specific clusters found in (B); and purple-colored are reaction clusters shown in both cases. Here, we found that reaction clusters 9 and 14 were regulated in a liver-specific manner at the levels of co-expression (B) and co-regulation (D) but were deregulated in HCC tumor tissues (E). Reaction clusters 9 and 14 included reactions associated with fatty acid synthesis, including glucose uptake, pyruvate synthesis, and citrate transport (F).

specifically in the liver. We first identified the group of metabolic reactions catalyzed by highly co-expressed enzymes in liver tissue by using the Human Metabolic Reaction database (HMR2; Mardinoglu *et al*, 2014a). Taking the maximal co-expression of enzymes between two metabolic reactions, we established a reaction co-expression matrix among all reactions (Fig 2A). From this matrix, we identified hundreds of clusters of reactions with highly co-expressed enzymes by using hierarchical clustering (Dataset EV7). By comparing their co-expression in liver tissue with the co-expression in adipose subcutaneous and skeletal muscle tissues, we identified metabolic reaction clusters that were co-expressed only in liver tissue (Fig 2B). On the basis of Fisher Z-transformed differences in tissue co-expression, we identified ten reaction clusters of the most differential co-expression levels between liver and muscle tissue (left, Fig 2B) or between liver and adipose tissue (right, Fig 2B); four reaction clusters overlapped in both cases.

Among those clusters, we determined the TFs that were highly co-bound with enzymes of those reaction clusters by using the hypergeometric test ($P < 0.05$) and found that each reaction cluster was governed by different sets of TFs (Fig 2C and Dataset EV8). For example, metabolic nuclear receptors, such as the farnesoid X receptor (FXR or NR1H4), pregnane X receptor (PXR or NR1I2), and RXRB, were highly co-bound with enzymes catalyzing the reaction in cluster 14, whereas SREBF2, a regulator of lipid homeostasis, was highly co-bound with enzymes catalyzing reactions in cluster 9. These findings indicated that enzymes catalyzing reactions in clusters 14 and 9 would share regulatory controls in response to metabolic alterations. Next, we determined which reaction clusters were enriched in influential TFs that we had identified. Using variable importance scores of TFs (Dataset EV6), we determined whether highly co-bound TFs of reaction clusters (Dataset EV8) had variable importance scores higher than the overall scores (KS one-sided test, $P < 0.25$; Fig 2D). We found that reaction clusters including 9, 14, 40, and 58 had significantly enriched TFs with highly variable importance scores, thus providing strong evidence of liver-specific regulation from physical and co-expression links. Through metabolic subsystem annotation from HMR2, we observed that reaction

cluster 9 was enriched in mitochondrial transport, pyruvate metabolism, and lipid metabolism, cluster 14 was enriched in fatty acid synthesis, cluster 40 was enriched in cholesterol metabolism, and cluster 58 was enriched in amino acid metabolism (hypergeometric test $P < 0.01$; Dataset EV9); these results indicated reactions involved in pathways that are regulated in only the liver and are primarily associated with lipid metabolism.

Next, we determined which liver reaction clusters were deregulated in HCC (Dataset EV7), on the basis of co-expression of those clusters in HCC tumor tissue from The Cancer Genome Atlas (TCGA; Fig 2E). We first retrieved RNA-seq data for 371 HCC tumors from TCGA and calculated Pearson's correlation coefficients between the gene pairs. As we identified liver-specific co-expression clusters (Fig 2B), we identified 10 reaction clusters that were deregulated in HCC tumors on the basis of Fisher Z-transformed differences in co-expression between GTEx liver data and TCGA HCC data (yellow and purple points in Fig 2E). Among those deregulated clusters, five reaction clusters were identified as liver-specific clusters as opposed to adipose and/or muscle tissue clusters (purple points in Fig 2E): reaction clusters 9, 14, 56, 62, and 65. Of note, reaction clusters 9 and 14 were identified on the basis of their regulation with strong evidence regarding co-expression (Fig 2B) and physical (Fig 2D) clues. In HCC, deregulation of these liver-specific reaction clusters primarily associated with fatty acid synthesis may be linked to HCC pathogenesis (Fig 2F).

## Identification of liver-specific FASN inhibitors for the treatment of NAFLD and HCC

The expression of fatty acid synthase (FASN), which catalyzes the last step in *de novo* lipogenesis (DNL), is significantly upregulated in NAFLD (Dorn *et al*, 2010) and HCC (Björnson *et al*, 2015). We have recently shown that short-term intervention with an isocaloric carbohydrate-restricted diet causes a large decrease in liver fat accompanied by striking rapid metabolic improvements (Mardinoglu *et al*, in preparation). We measured clinical characteristics, body composition, liver fat, hepatic DNL, and hepatic

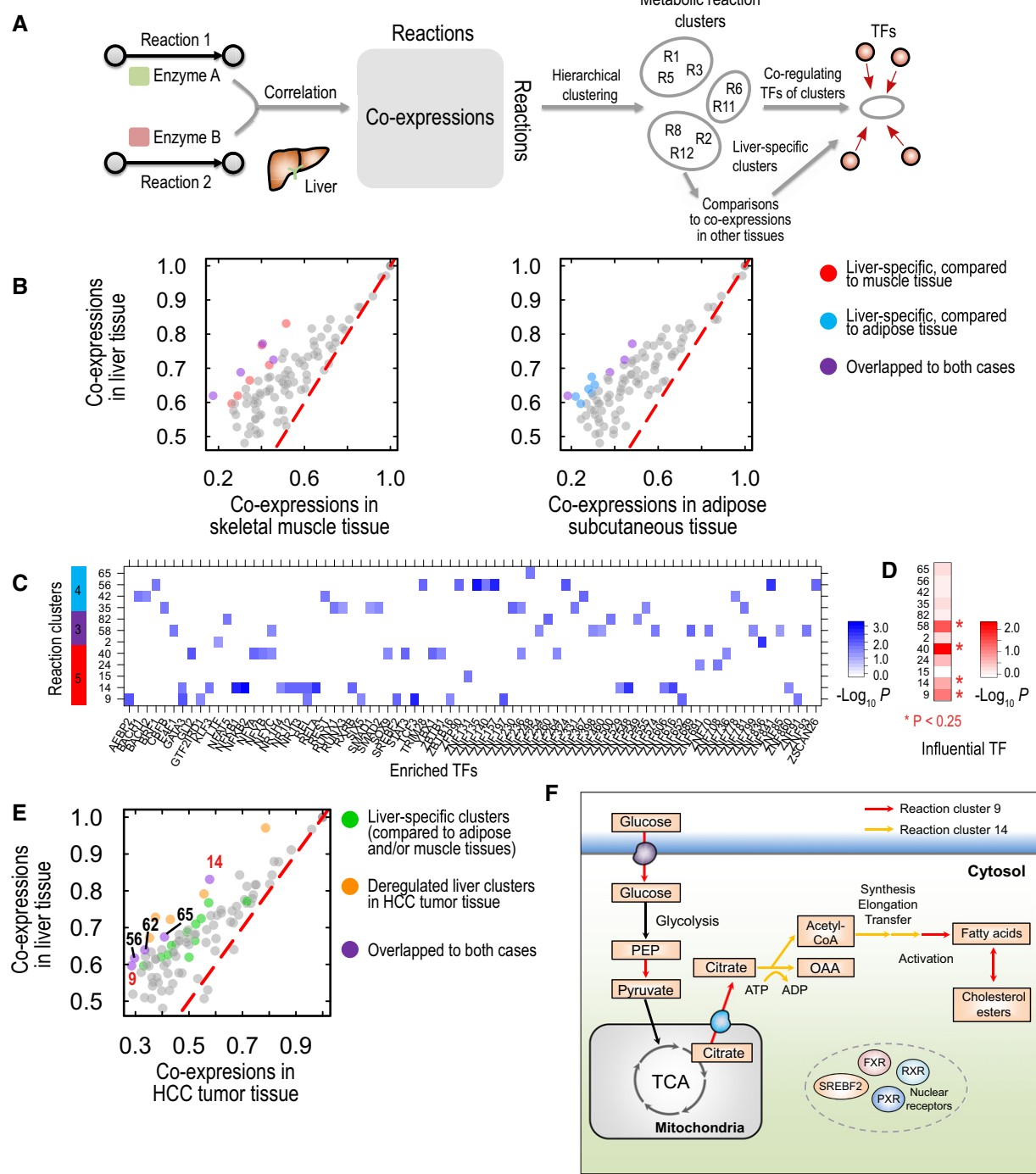

**Figure 2.**

<br>

β-oxidation and found that the diet caused rapid and significant sustained decreases in liver fat, DNL, plasma triglycerides, and very low-density lipoprotein triglycerides and a parallel increase in β-hydroxybutyrate, an indicator of increased hepatic β-oxidation. Hence, DNL may be targeted for the development of effective treatment strategies for NAFLD and other chronic liver diseases, for example, HCC.

However, small-molecule FASN inhibitors (e.g., C75, cerulenin) suffer from pharmacological limitations that prevent their

development as systemic drugs (Pandey *et al*, 2012). These side effects can also be explained by the high expression of FASN in almost all major human tissues (Uhlen *et al*, 2015, 2016). Thus, we hypothesized that the identification of liver-specific FASN inhibitors might allow for the development of effective treatment strategies for NAFLD and HCC. We identified highly co-expressed genes with FASN on the basis of CNs generated using GTEx and TCGA data, which have been used as representative datasets for NAFLD (Dataset EV10) and HCC (Dataset EV11), respectively. Our CN

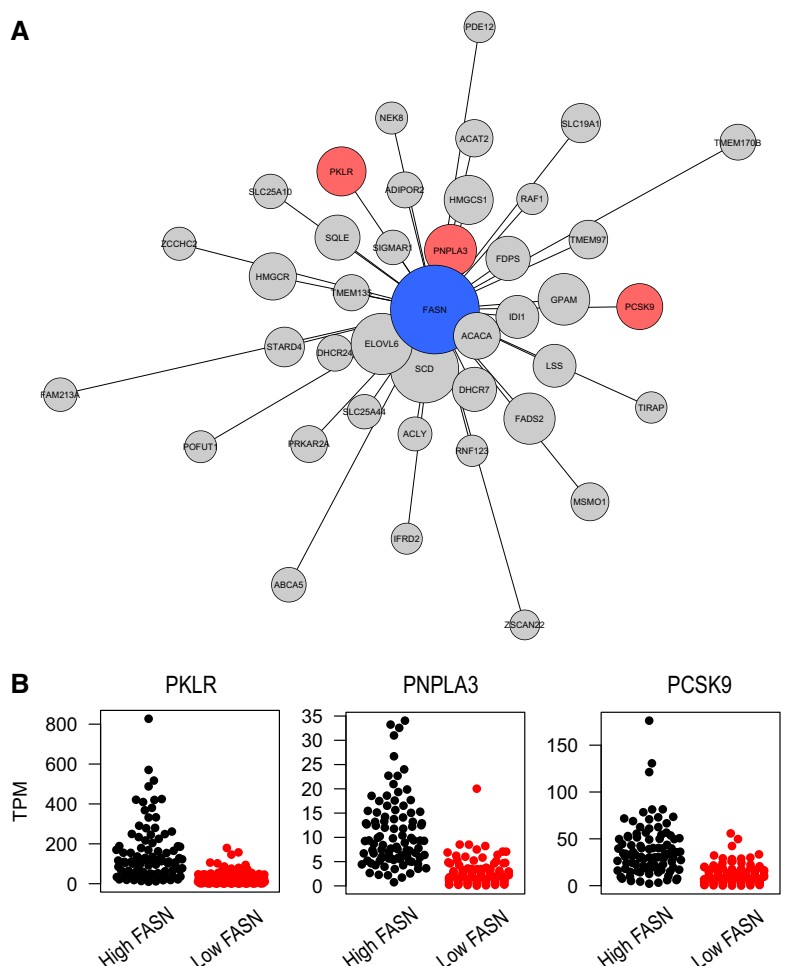

**Figure 3.  FASN CN in HCC tumor tissue.**

A  We present the 40 genes with the highest co-expression with FASN in HCC tumor tissue based on both log-transformed and raw expression values (Datasets EV11 and EV12). The lengths of edges were inversely proportional to the co-expression of corresponding gene pairs; thus, genes close to FASN were more co-expressed than others. Node sizes were inversely proportional to adjusted p-values of differential expressions between patients between high (upper quartile) and low (lower quartile) FASN expressions (Dataset EV13). We colored co-expressed genes on the basis of their liver specificity; red-colored genes were liver tissue-enriched (based on HPA ver. 16 annotation) and co-expressed with FASN in fewer than three human tissues (Dataset EV10); FASN alone was colored blue.

B  We show expressions of PKLR, PNPLA3, and PCSK9 in HCC patients with high and low FASN expressions. We found that liver-specific genes were significantly (adjusted $P < 1.0 \times 10^{-10}$) upregulated in patients high FASN expression compared to those with low FASN expression (Dataset EV13).

analysis allowed the identification of genes functionally related to FASN in liver and other major human tissues.

Using DAVID, we found that the top 100 genes co-expressed with FASN in liver CN (Dataset EV10) were significantly enriched in GO BP terms, including carboxylic acid, oxoacid, hexose, monosaccharide, acyl-CoA and fatty acid metabolic processes, protein transport, secretion, regulation of secretion, and negative regulation of cell communication (Huang *et al*, 2009). We also compared the top 100 co-expressed genes with FASN in liver CN (Dataset EV10) with the genes in 45 other tissue CNs (Dataset EV10) and identified pyruvate kinase liver and red blood cell (PKLR), an enzyme phosphorylating pyruvate from glycolysis to the TCA (citric acid) cycle and also in fatty acid synthesis; PKLR is also shown in Fig 2F, as a liver-specific gene co-expressed with FASN.

In HCC CN, we found that the top 100 genes co-expressed with FASN (Dataset EV11) were significantly enriched in GO BP

terms, including animal organ development, carboxylic acid, oxoacid, cholesterol, secondary alcohol, sterol and fatty acid metabolic processes, along with steroid, sterol, and alcohol biosynthetic processes. We also found ELOVL6, ACACA, and SCD, involved in fatty acid biosynthesis, as the top genes co-expressed with FASN in HCC CN (Dataset EV11). In addition, we calculated Pearson's correlations from log-transformed gene expressions in HCC in order to check for robustly co-expressed genes (Dataset EV12). We found that 40 of the genes in top-100 co-expressed genes were significantly co-expressed with FASN before (Dataset EV11) and after log transformation (Dataset EV12). We compared those robustly co-expressed genes with FASN in HCC CN (Dataset EV12) with the top 100 co-expressed genes in CNs of 46 human tissues (Dataset EV10) and found PKLR, patatin-like phospholipase domain containing 3 (PNPLA3), and proprotein convertase subtilisin/kexin type 9 (PCSK9),

    

referred to as liver-specific genes, as the only genes co-expressed with FASN in less than three human tissues (Fig 3A).

Next, we compared the global gene expression profiling of the HCC tumors in the upper quartile with highest expression of FASN ($n = 93$) with the lower quartile with the lowest expression of FASN ($n = 93$) using DESeq package (Anders & Huber, 2010) to analyze the expression profile of liver-specific genes and their key role in cancer progression. We found that the expression of PKLR, PNPLA3, and PCSK9 was significantly increased in patients with high FASN expression compared to those with low FASN expression (Fig 3B and Dataset EV13).

We also determined the expression patterns of PKLR, PCSK9 and PNPLA3 in the Human Protein Atlas, in which the expression of all human protein coding genes have been measured in 32 major human tissues, and these genes were identified as liver-specific genes based on protein and mRNA expression (Kampf *et al*, 2014; Uhlen *et al*, 2015). Hence, these liver-specific genes, including PKLR, PNPLA3, and PCSK9, may potentially be targeted for the treatment of HCC, and PKLR may potentially be targeted for the treatment of NAFLD. Due to the direct involvement of PKLR, PNPLA3, and PCSK9 in lipid metabolism, we focused on the relationship between FASN and these three liver-specific genes in the rest of our studies.

## Validation of physical interactions by using human cancer cell lines

We hypothesized that inhibition of liver-specific targets that are co-expressed with FASN would inhibit FASN expression and decrease fat synthesis. This inhibition would also inhibit tumor growth in the case of HCC, because fatty acids play key roles in HCC progression and development. To validate our hypothesis and to demonstrate the physical interactions between FASN and liver-specific genes, we first screened the cell lines with the highest PKLR mRNA expression levels by using the data generated in the Human Protein Atlas (Uhlen *et al*, 2015). We identified the K562 leukemia cell line as having the highest PKLR expression using our recently published data in Cell Atlas (Thul *et al*, 2017) and treated the cell line with C75, a FASN inhibitor, at different concentrations for 24 h. The purpose of this experiment was to test whether FASN inhibition may affect the co-expressed genes with FASN (i.e., PKLR).

We found that FASN and PKLR expression levels were significantly decreased (Fig 4A). Moreover, we determined that decreased FASN and PKLR expression resulted in a significant decrease in cell growth (Fig 4B). Next, we treated the HepG2 human cancer cell line with C75 and found that FASN expression and the expression levels of liver-specific genes, including PKLR, PNPLA3, and PCSK9, were significantly decreased after 24 h (Fig 4C). Similarly, we found that the growth of HepG2 cells was significantly decreased after treatment of the cells with different concentrations of C75 (Fig 4D).

However, during the treatment of the cells with C75, other metabolic pathways may also have been regulated, and gene expression may have been affected by nonspecific binding of C75. In this context, we used siRNA to decrease PKLR expression in the K562 cell line and observed that FASN expression (Fig 4E) and cell growth (Fig 4F) were significantly decreased, similarly to our observations after treating the cells with C75. Moreover, we inhibited PKLR expression in the HepG2 cell line and found that FASN expression (Fig 4G) and cell growth (Fig 4H) were significantly decreased, as expected.

## Validation of physical interactions in mice and human

To show the functional relationship between FASN and the liver-specific targets identified here *in vivo*, we fed mice (C57Bl/6N) a zero-fat high-sucrose diet (HSD) for 2 weeks that induced the development of fatty livers in mice (Fig 5A). We collected liver tissue samples from the mice fed a HSD and compared the expression profiles of the genes in the liver of these mice with those in the liver of mice fed a chow diet (CD). We observed that liver TG content in the mice was significantly increased in the HSD fed group compared with the control (Fig 5A). Next, we determined the expression profiles of FASN and our liver-specific gene targets and found that their expression levels were significantly increased in mice fed the HSD compared with mice fed the CD in parallel with the increase in liver fat (Fig 5B).

It has been reported that circulating PCSK9 levels increase with the severity of hepatic fat accumulation in patients at risk of NASH and PCSK9 mRNA levels in liver have been linked with steatosis severity (Ruscica *et al*, 2016). In this context, we fed the wild-type (WT) and PCSK9 knockout (KO) mice a CD for 10 weeks. We collected liver tissue samples from the WT and PCSK9 KO mice and compared the expression levels of FASN and the other identified gene targets. We determined the expression levels of these genes in PCSK9 KO mice and found that they were significantly downregulated (Fig 5C). Hence, our analysis suggested to targeting of PCSK9 for the development of efficient treatment strategies for HCC patients.

Endocannabinoids acting on the hepatic cannabinoid-1 receptor ($CB_1R$) promote DNL by increasing the expression of genes involved in lipid metabolism including FASN, SREBF1, and acetyl-CoA carboxylase-1 (ACACA) (Osei-Hyiaman *et al*, 2005). $CB_1R$ has been implicated in the pathology of different liver diseases with various etiologies including NAFLD (Osei-Hyiaman *et al*, 2008), AFLD (Jeong *et al*, 2008), viral hepatitis (Hezode *et al*, 2005), liver fibrosis (Teixeira-Clerc *et al*, 2006), cirrhosis (Giannone *et al*, 2012), and liver cancer (Mukhopadhyay *et al*, 2015). Activation of the endocannabinoid/$CB_1R$ system inhibits fatty acid β-oxidation in the liver (Osei-Hyiaman *et al*, 2008), interrupts hepatic carbohydrate and cholesterol metabolism (Jourdan *et al*, 2012), and contributes to diet-induced obesity and NAFLD. It has been observed that activation of hepatic $CB_1R$ promoted the initiation and progression of chemically induced HCC in mice (Mukhopadhyay *et al*, 2015). Considering the associations between $CB_1R$, liver fibrosis, and HCC even in the absence of obesity, we analyzed the expression of Fasn as well as liver-specific gene targets in animal models of liver cancer. We measured the expression of the Fasn and the liver-specific gene targets including Pklr, Pcsk9, and Pnpla3 and found that their expression was significantly ($P < 0.05$) increased in HCC tumor compared to adjacent noncancerous samples obtained from $CB_1R^{+/+}$ mice (Fig 5D), with much smaller increases noted in corresponding samples $CB_1R^{-/-}$ mice. When we measured the expression of these genes in HCC tumor and noncancerous samples obtained from $CB_1R^{-/-}$ mice, we found that the increase in the expression of Fasn as well as liver-specific gene targets is attenuated in conjunction with the decrease in tumor growth $CB_1R^{-/-}$ mice (Fig 5E).

We finally validated our predictions by treating primary human hepatocytes with C75 and found that the expression levels of FASN PKLR, PCSK9 and PNPLA3 were significantly decreased (Fig 5F). Liver diseases connected to FASN can thus be treated (by silencing

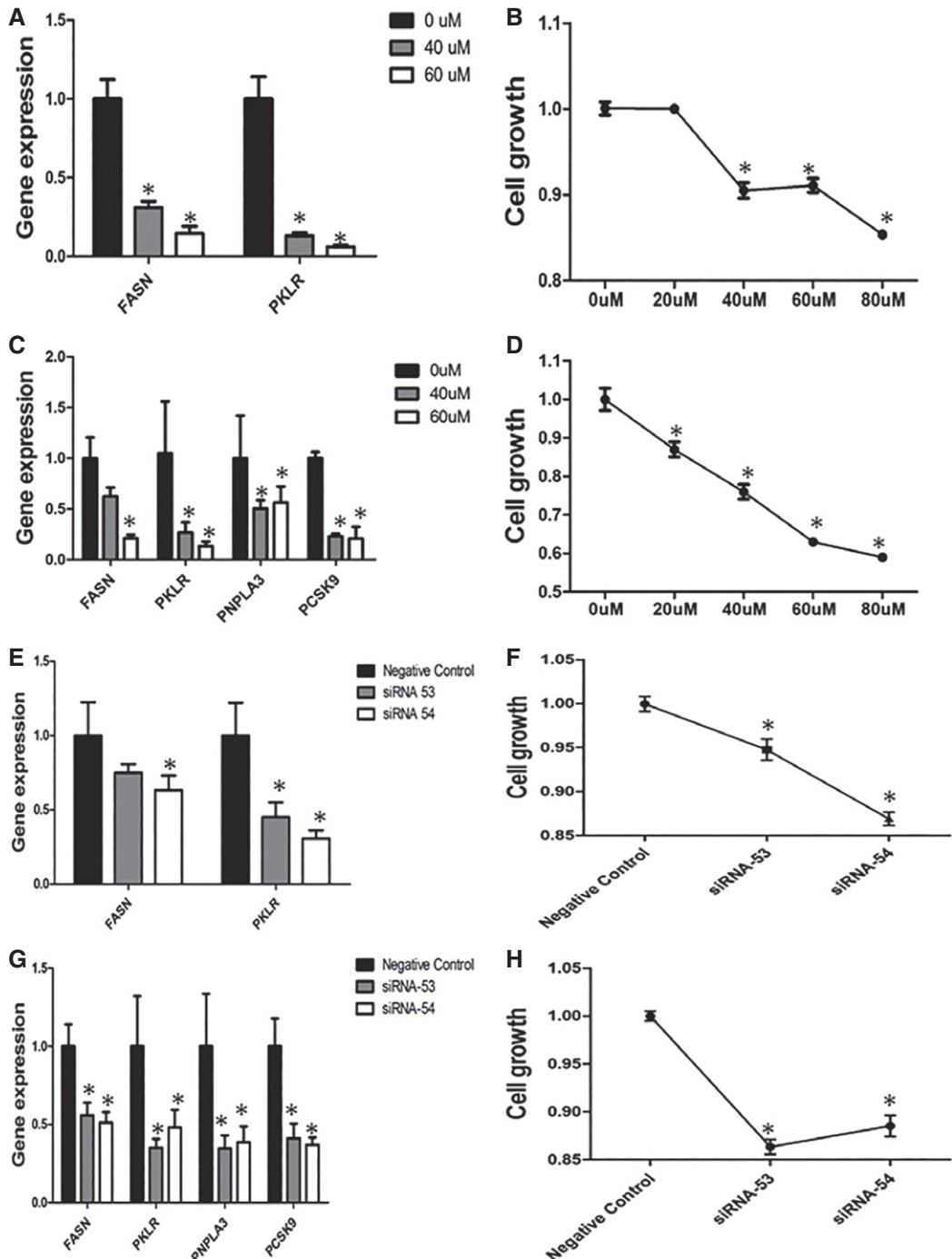

**Figure 4.   Gene expression and proliferation of K562 and HepG2 cells after interference by C75 and PKLR-specific siRNA.**

A   FASN and PKLR expression levels in K562 cells after interference by different doses (0, 40, and 60 μM) of C75.
B   Cell growth in K562 cells after interference by different doses (0, 20, 40, 60, and 80 μM) of C75.
C   FASN, PKLR, PNPLA3, and PCSK9 expression levels in HepG2 cells after interference by different doses (0, 40, and 60 μM) of C75.
D   Cell growth in HepG2 cells after interference by different doses (0, 20, 40, 60, and 80 μM) of C75.
E   FASN and PKLR expression levels in K562 cells after interference by PKLR-specific siRNA (siRNA 53, siRNA 54).
F   Cell growth in K562 cells after interference by PKLR-specific siRNA (siRNA 53, siRNA 54).
G   FASN, PKLR, PNPLA3, and PCSK9 expression levels in HepG2 cells after interference by PKLR-specific siRNA (siRNA 53, siRNA 54).
H   Cell growth in HepG2 cells after interference by PKLR-specific siRNA (siRNA 53, siRNA 54).

Data information: RNA was isolated for RT–PCR after interference for 24 h; GAPDH was set as the internal reference. Cell counting was performed after interference for 72 h. Data are presented as the means ± standard errors of five independent experiments. Comparisons were performed by one-way ANOVA. Samples without any interference were assigned as controls. * represents a significant difference compared with the value in the control group ($P < 0.05$).

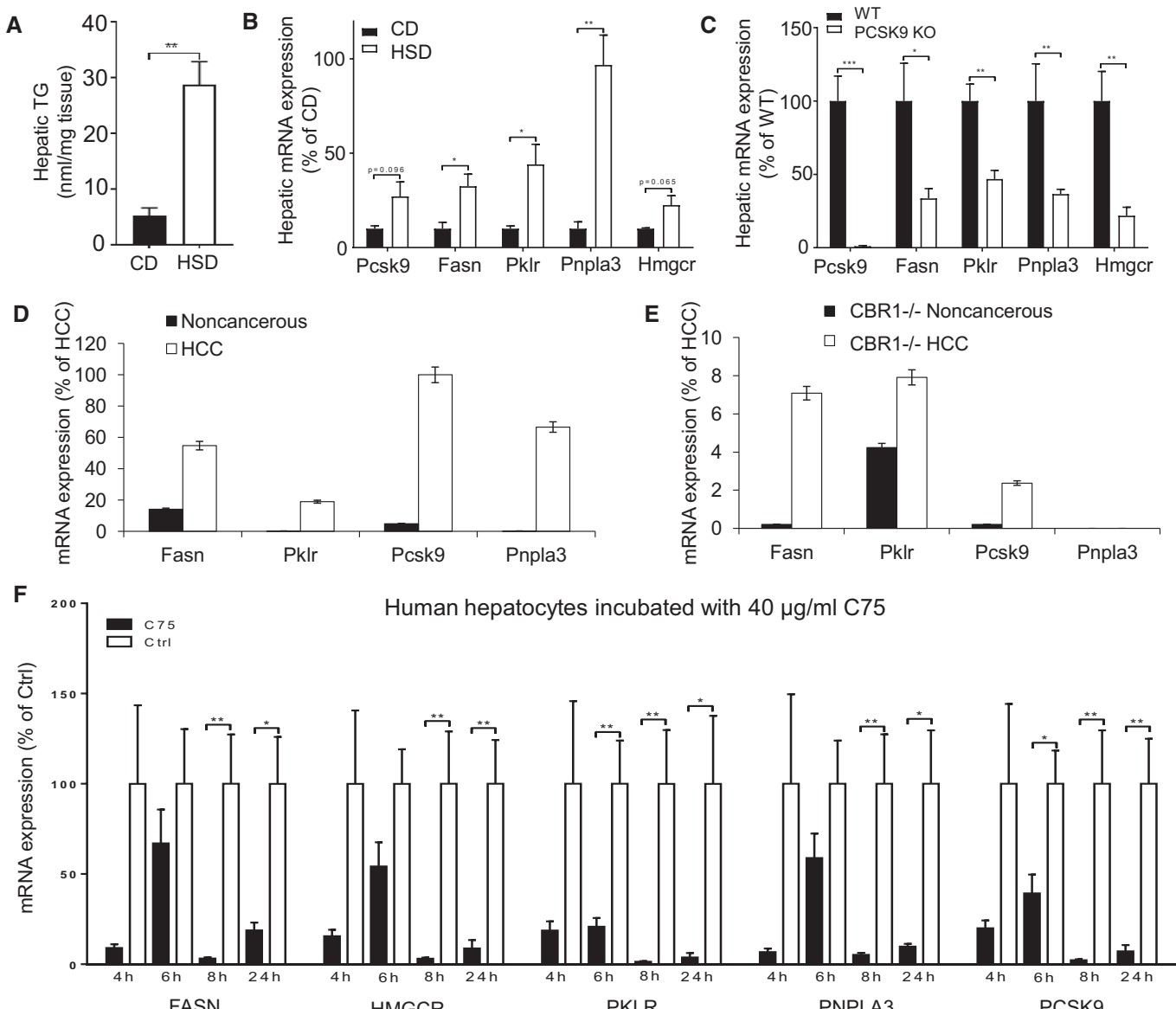

**Figure 5.  The relationship between the genes in mice and human samples.**

A    TG content in the liver tissue of mice fed a zero-fat high-sucrose diet (HSD) and chow diet (CD) for 2 weeks (*n* = 10).

B–F  The hepatic mRNA expressions of the Fasn, Pklr, Pcsk9, Pnpla3, and Hmgcr is measured in (B) mice fed a HSD and CD for 2 weeks, (C) Pcsk9 knockout and its littermates (WT) fed a CD for 10 weeks, (D) wild-type HCC tumor and noncancerous samples, (E) CB₁R knockout HCC tumor and noncancerous samples, and (F) primary human hepatocytes treated with C75 for 4, 6, 8, and 24 h.

Data information: Data are presented as the means ± standard errors of independent experiments.  Student's *t*-test; *$P < 0.05$, **$P < 0.01$, ***$P < 0.001$ represents a significant difference compared with the value in the control group.

PKLR, PCSK9 or PNPLA3) without experiencing the side effects associated with direct FASN inhibition.

## Discussion

Rapid advances in omics technologies along with the adoption of large shared public databases have allowed for the generation and aggregation of massive sample datasets that can be used to construct comprehensive biological networks. These networks may provide a

scaffold for the integration of omics data, thereby revealing the underlying molecular mechanisms involved in disease appearance and providing a better understanding of the variations in healthy and diseased tissue that may be used in the development of effective treatment strategies. In this study, we generated tissue-specific CNs for 46 major human tissues and human liver cancer and explored the tissue-specific functions by using the topologies provided by these networks. An important aspect of a gene CN is modularity: Genes that are highly interconnected within the network are usually involved in the same biological modules or pathways. We compared

the tissue-specific CNs with the recently generated INs and found that physical interactions revealed meaningful functional relationships between genes. Next, using CNs, we investigated the emergent properties and behaviors of the affected genes in response to NAFLD and HCC at the system level rather than focusing on their individual functions and clinical utilities. The use of CNs also allowed us to obtain detailed information about the systems-level properties of such complex liver diseases.

Correlation analysis is used to identify co-expressions between different genes based on mRNA expression data. Although co-expression does not necessarily indicate a relationship among transcript levels, functional relationships between encoded protein-coding genes have been shown in our study. Co-expression analysis is applied to identify important alterations related to lipid metabolism, and promotion of cell survival and cell growth. In order to avoid possible false positives from co-expressed genes, we applied strict cutoff of Pearson's correlation coefficients, but it would be necessary to develop advanced method to identify co-expressed genes for smaller number of false positives. Here, we calculated correlations of gene expressions using normal tissue samples obtained from general population (i.e., GTEx) without selection bias, thus expected to identify general functional relationships, not only associated with a specific clinical indication. However, it is noteworthy that RNA-seq data generated for large and extensively phenotyped cohorts would improve our understanding of functional relationships between genes in a specific clinical indication.

Considering the increased prevalence of NAFLD in the worldwide as a hidden epidemic (Younossi *et al*, 2016), it is not implausible to predict that NAFLD may become responsible for the future clinical and economical burden of HCC (Younossi *et al*, 2015). It is also reported that HCC patients with NAFLD have poor outcome for disease progression (Younossi *et al*, 2015). Therefore, it is important to identify common biological pathways and gene targets as driving forces in the pathologies for developing effective therapeutic modalities. In this context, system biology may contribute to identification of novel targets that can be used in the development of efficient treatment strategies. Our recent analysis described in Mardinoglu *et al* (in preparation) and previous analysis (Mardinoglu *et al*, 2014a) indicated that DNL is a key pathway involved in the progression of the NAFLD. Previous analysis has also revealed DNL as one of the most tightly regulated pathways in HCC compared with noncancerous liver tissue (Björnson *et al*, 2015). FASN (rate-limiting enzyme in DNL) is believed to play key roles in NAFLD progression and development and HCC; therefore, inhibiting FASN may downregulate the DNL, decrease the accumulated fat within the cells, and decrease tumor growth. To improve the success rate of NAFLD and HCC treatment and to improve the survival prognosis of HCC patients, identification of nontoxic FASN inhibitors is required. To identify possible inhibitors, correlation analysis was applied to mRNA expression data derived from liver tissue with varying degree of fat and tumor samples. Co-expression analysis in these samples revealed that FASN is co-expressed with a number of genes that play roles in crucial biological processes involved in fat accumulation and cancer cell metabolism.

We analyzed the majority of publicly available liver tissue gene expression data, performed correlation analysis, and ultimately generated co-expression networks for FASN. We identified a number of liver-specific gene targets that can be inhibited with chemical compounds or monoclonal antibodies including PKLR, PCSK9, and PNPLA3, for effective treatment of chronic liver diseases. We finally validated our predictions by demonstrating the functional relationships among the expression of these genes and FASN, liver fat, and cell growth, by using human cancer cell lines, mouse liver samples (four different mouse studies), and human hepatocytes.

Wang *et al* (2002) have provided evidence that variants in the PKLR gene are associated with an increased risk of T2D, which has a pathogenesis similar to that of NAFLD. Ruscica *et al* (2016) have associated circulating PCSK9 levels with accumulated liver fat. In 201 consecutive patients biopsied for suspected NASH, liver damage has been quantified by NAFLD activity score, circulating PCSK9 by ELISA, and hepatic mRNA by qRT–PCR in 76 of the patients. Circulating PCSK9 has been found to be significantly associated with hepatic steatosis grade, necroinflammation, ballooning, and fibrosis stage (Ruscica *et al*, 2016). Circulating PCSK9 has also been found to be significantly associated with hepatic expression of SREBP-1c and FASN, whereas PCSK9 mRNA levels have been found to be significantly correlated with steatosis severity and hepatic APOB, SREBP-1c, and FASN expression (Ruscica *et al*, 2016). Aragones *et al* (2016) have evaluated the association between liver PNPLA3 expression, key genes in lipid metabolism, and the presence of NAFLD in morbidly obese women and have reported that PNPLA3 expression was related to HS in these subjects. Their analysis indicates that PNPLA3 may be related to lipid accumulation in the liver, mainly in the development and progression of simple steatosis. PNPLA3 was also emphasized as a genetic determinant of risk factor for the severity of NAFLD (Salameh *et al*, 2016). Furthermore, higher prevalence for HCC development and poorer prognosis was reported to be associated with PNPLA3 polymorphism in viral and nonviral chronic liver diseases (Khlaiphuengsin *et al*, 2015). A potential unifying factor upstream of these genes is the cannabinoid-1 receptor, stimulation of which was found to upregulate several of the above-listed target proteins, including Fasn, Pklr, Pnpla3, and Pcsk9 in mouse models of obesity/metabolic syndrome and HCC, as documented and detailed.

Moreover, we found a number of genes, for example, ACACA, were significantly co-expressed with FASN. It has been suggested that inhibition of ACACA may be useful in treating a variety of metabolic disorders, including metabolic syndrome, type 2 diabetes mellitus, and fatty liver disease (Harriman *et al*, 2016). However, our analysis indicated that potential inhibition of ACACA may have severe side effects in other human tissues as the inhibitors of FASN.

In conclusion, we demonstrated a strategy whereby tissue-specific CNs can be used to identify deregulations of biological functions in response to disease and reveal the effects on relevant expression of genes in liver. Eventually, we identified liver-specific drug targets that can be used in effective treatment of liver diseases including NAFLD and HCC.

## Materials and Methods

### Tissue-specific CNs

RNA-seq data from human tissues were downloaded from the GTEx database, and their reads per kilobase per million (RPKM) values were transformed into transcripts per kilobase per million (TPM)

values. From each RNA-seq dataset, we excluded one-third of the genes with the lowest expression levels from calculating Pearson's correlation coefficients of gene expression and combined the highest correlated gene pairs into a respective tissue CN.

On the basis of the network connectivity of each tissue CN, we clustered co-expressed genes by using the modularity-based random walk method from the cluster walktrap function of the igraph package in R (Pons & Latapy, 2005). Among those gene groups, we selected the half of the highest connected groups as key co-expression clusters, on the basis of their clustering coefficients. We produced a liver CN with those co-expression clusters as nodes and their significant connections by edges (Fig 1B). For example, the edges of two co-expression clusters, A and B, were identified if their observed connections ($O_{AB}$) were twice as high as the expected connections ($E_{AB}$). Expected connections were defined by the normalized sum of multiplications of the degree of connectivity of genes in two clusters (i.e., $E_{AB} = \Sigma\, k_a \times k_b/2N$; $a \in A$, $b \in$, $N$ = all edges in the network). Next, we collected genes belonging to key co-expression clusters in each tissue and tested whether they were enriched in biological process GO terms in MSigDB (Subramanian et al, 2005) by using hypergeometric tests.

**Comparing physical networks with CNs**

We used RNs and PPINs of liver, skeletal muscle, and adipose tissues from our prior published data (Lee et al, 2016) as sources of physical interaction network data. We identified co-regulated gene pairs from the physical interactions by selecting genes sharing TF binding (from RNs) or genes sharing protein interactions (from PPINs); on the basis of the number of co-bound TFs or co-interacting proteins, called co-regulators, we selected the highest co-regulated gene pairs (i.e., gene pairs with the highest numbers of co-bound TFs or co-interacting proteins; top 0.1%) and gene pairs with no co-regulation. Using Pearson's correlation coefficients of the gene expression level of each tissue from GTEx, we calculated the mean Pearson's correlation coefficients of gene pairs of interest, such as physically linked gene pairs or co-regulated gene pairs. The edges of physical networks or their co-regulatory networks were randomly permutated among genes in respective actual networks by 1,000×, and their mean correlation coefficients were compared with those from original networks. We also identified TFs or proteins highly co-expressed with respective bound genes by comparing the co-expression levels of given gene pairs to overall levels (Datasets EV4 and EV5). We selected TFs or proteins if co-expression levels of given linked gene pairs were higher than the overall levels by using Kolmogorov–Smirnov (KS) tests ($P < 0.05$) and absolute values of mean Pearson's correlation coefficients ($> 0.1$). Finally, we examined mean co-expression levels of co-regulated gene pairs according to their number of co-regulators. Increasing the cutoff number of co-regulators, we selected co-regulated gene pairs exceeding the cutoff and calculated mean Pearson's correlation coefficients of the corresponding gene pairs.

**Finding the most influential TFs for co-expression on the basis of variable importance score**

For liver tissue, we constructed a feature matrix between the most highly co-expressed gene pairs (top 1%) and their co-bound TFs. In this matrix, we assigned a value of 1 for given co-expressed gene pairs that were co-bound by TFs and zero otherwise. For each co-expressed gene pair, we considered co-bound TFs as predictor variables and the co-expression value as the response variable and fitted them to the Random Forest model. From the model, we calculated variable importance scores of all TFs and used them as a metric to show the most influential TFs for co-expression.

**Identifying liver-specific reaction clusters on the basis of tissue co-expression of enzymes**

From the HMR2 database, we collected human metabolic reactions with known enzymes. Between the two reactions, we calculated Pearson's correlation coefficients of enzyme gene expression in liver tissue, and if there were multiple enzymes for a single reaction, we took the maximum value among possible co-expressions. On the basis of co-expression values among metabolic reactions, we performed hierarchical clustering and classified reactions into 100 clusters for liver tissue. To compare the co-expression values of metabolic reactions in different tissues and tumor tissue, we calculated the co-expression values in not only liver tissue but also adipose subcutaneous, skeletal muscle, and HCC tumor tissues. Subsequently, we identified the differences in mean co-expression values of given reaction clusters between liver tissue and other tissues after transforming co-expression values into Fisher Z-values. On the basis of differential Fisher Z-values, we took the top 1% of reaction clusters of the highest in each comparison (adipose or muscle tissues) and identified them as liver-specific reaction clusters. Likewise, we identified an HCC-deregulated reaction cluster on the basis of differential Fisher Z-values (top 1%).

Finally, using liver RN, we examined TFs highly bound at genes encoding enzymes in given liver reaction clusters and identified TFs whose binding was significantly enriched in given clusters by hypergeometric tests. From those enriched TFs, we selected reaction clusters in which the enriched TFs had higher variable importance scores than overall scores according to KS tests ($P < 0.25$) and denoted them as highly regulated reaction clusters in liver tissue.

**FASN co-expressed genes in human tissues and HCC tumors**

In each normal tissue (GTEx) or HCC tumor tissue (TCGA), we calculated Pearson's correlation coefficients of gene expression between FASN and other protein-coding genes expressing more than 1 TPM and selected the top 100 most correlated genes (Datasets EV10 and EV11). In addition, we calculated correlations of log-transformed expression values in HCC tumors (Dataset EV12). To select tissue-specific genes, we examined the top 100 most correlated genes to FASN in HCC tumor tissue by the RNA tissue category in the Human Protein Atlas (ver. 16); in particular, genes "tissue-enhanced" or "tissue-enriched" in liver tissue were selected (Fig 3). We also examined those genes with tissues that were most correlated (Dataset EV10) and selected genes that were present in fewer than three human tissues.

**Differential expression analysis of HCC patients stratified based on FASN expressions**

Using raw count data from HCC patients, we stratified patients into two groups: patients having FASN expression above the upper

quartile and patients having FASN expression under the lower quartile. Between the two groups, we examined differentially expressed genes by negative binomial test using DESeq (Anders & Huber, 2010; Dataset EV13).

## Cell line experiments

For subsequent experiments, we selected K562 and HepG2, human immortalized myelogenous leukemia and hepatic cell lines, respectively. Both cell lines were cultured in RPMI-1640 medium (R2405; Sigma-Aldrich) supplemented with 10% fetal bovine serum (FBS, F2442; Sigma-Aldrich) and incubated in 5% $CO_2$ humidity at 37°C.

To confirm the speculated CN, we chose chemical inhibitors and RNA interference (RNAi) assays to interfere with the immortalized cell lines (including K562 and HepG2) and observed the subsequent candidate gene expression and cell growth patterns. More specifically, the experimental protocol was as follows: (i) C75 (C5490; Sigma-Aldrich), as a well-known fatty acid synthase [FAS] inhibitor, was added to cells at 80% confluence with a final concentration of 20, 40, 60, or 80 μM (taking cells without C75 interference as the control); and (ii) cells at 80% confluence were separately transfected with three pairs of Silencer® pre-designed PKLR-targeted siRNAs (clone ID: 53, 54; Life Technologies; Dataset EV14) at 15 nM by using Lipofectamine® RNAiMAX (13778075; Life Technologies). Cells incubated in medium with nontarget negative control siRNA at 15 nM (4390843; Life Technologies) were assigned as the control.

Total RNA was isolated with TRIzol reagent (15596026, Thermo Fisher Scientific) after treatment with C75 or siRNA for 24 h. The expression profiles of key genes (FASN, PKLR in K562/HepG2, and PNPLA3, PCSK9 only in HepG2 cells) in the co-expression network were measured and analyzed via quantitative real-time PCR with iTaq Universal SYBR Green One-Step Kit (1725151; Bio-Rad), using anchored oligo (dT) primer based on CFX96™ detection system (Bio-Rad). GAPDH was set as the internal control for normalization, and the primer sequences are listed in Dataset EV15. Variation in cell proliferation was detected with a Cell Counting Kit-8 (CCK-8, CK04; Dojindo) after interference by C75/siRNA for 72 h. All experiments were performed strictly according to the manufacturer's instructions and were repeated at least in triplicate for three samples and yielded similar results.

## Mouse experiments

Twenty male C57BL/6N mice were fed a standard mouse chow diet (Purina 7012; Harlan Teklad) and housed under a 12-h light–dark cycle. From 8 weeks of age, the mice were fed either a HSD diet (TD.88137; Harlan Laboratories, WI, USA) or CD for 2 weeks. The mice were housed at the University of Gothenburg animal facility (Lab of Exp Biomed) and supervised by university veterinarians and professional staff. The health status of our mice is constantly monitored according to the rules established by the Federation of European Laboratory Animal Science Associations. The experiments were approved by the Gothenburg Ethical Committee on Animal Experiments.

In liver cancer mouse model, we injected 25 mg/kg of DEN (Sigma) to $CB_1R^{+/+}$ and $CB_1R^{-/-}$ littermates in C57BL/6J

background after 2 weeks of birth, verified the presence of the HCC tumor, and measured its size with magnetic resonance imaging (MRI) 8 months after the DEN administration (Mukhopadhyay *et al*, 2015). It has been observed that activation of hepatic $CB_1R$ promoted the initiation and progression of chemically induced HCC in mice (Mukhopadhyay *et al*, 2015). Total RNA was isolated from tumor area and noncancerous area of liver tissue samples obtained from six DEN-treated (HCC) $CB_1R^{+/+}$ and six $CB_1R^{-/-}$ mice (Mukhopadhyay *et al*, 2015). rRNA-depleted RNA, 100 ng for each sample, was treated with RNase III to generate 100- to 200-nt fragments, which were pooled and processed for RNA sequencing. All data were normalized based on housekeeping genes used by CLC Genomics Workbench program (version 5.1; CLC Bio, Boston, USA). These absolute numbers were extracted from the reads, and the data were adjusted to non-HCC biopsies for each gene.

## Human hepatocytes

Human primary hepatocytes were purchased from Biopredic International. Twenty-four hours after arrival (48 h after isolation), human hepatocytes were treated with 40 μg/ml of C75 or dimethyl sulfoxide (DMSO) for 4, 6, 8, or 24 h.

## mRNA expression in mouse liver and human primary hepatocytes

Total RNA was isolated from human hepatocytes and snap-frozen mouse liver with an RNeasy Mini Kit (Qiagen). cDNA was synthesized with a high-capacity cDNA Reverse Transcription Kit (Applied Biosystems) and random primers. The mRNA expression levels of genes of interest were analyzed via TaqMan real-time PCR in a ViiATM7 System (Applied Biosystems). The TaqMan Gene Expression assays used were Mm01263610_m1 (for mouse Pcsk9), Mm00662319_m1 (mouse Fasn), Mm00443090_m1 (mouse Pklr), Mm00504420_m1 (mouse Pnpla3), Mm01282499_m1 (mouse Hmgcr), Hs00545399_m1 (human PCSK9), Hs01005622_m1 (human FASN), Hs00176075_m1 (human PKLR), Hs00228747_m1 (human PNPLA3), and Hs00168352_m1 (human HMGCR) (all from Applied Biosystems). Hprt (mouse Mm03024075_m1) and GADPH (human Hs02758991_g1) (Applied Biosystems) were used as internal controls.

## Data availability

The networks used in this study are provided on an interactive web page at: http://inetmodels.com. Co-expression networks have also been uploaded to the NDEx (ndexbio.org), and network IDs are provided in Dataset EV1.

**Expanded View** for this article is available online.

## Acknowledgements

This work was financially supported by the Knut and Alice Wallenberg Foundation, Swedish Research Foundation, and EU Seventh Framework Programme RESOLVE. The research leading to these results received support from the Innovative Medicines Initiative Joint Undertaking under EMIF grant agreement no. 115372. The computations of network generations were performed using resources provided by the Swedish National Infrastructure for Computing (SNIC) at C3SE and UPPMAX.

## Author contributions

SL, CZ, and AM generated the co-expression networks and analyzed the clinical data, together with TK, MG, BDP, MSn, JN, and MU. ZL, MK, BM, MB, RC, NS, SD, SJE, FB, GK, MSt, JKP, AMH, US, and JB measured the expression levels of the genes in the cell lines, mouse liver samples, and human hepatocytes. SL, CZ, and AM wrote the manuscript, and all authors were involved in editing the manuscript.

## Conflict of interest

AM, JB, and MU have filed a patent application about the siRNA of targets reported in this study. The other authors declare no conflict of interests.

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
