## [Review Process File · Molecular Systems Biology]

Network analyses identify liver-specific targets for treating liver diseases

Sunjae Lee, Cheng Zhang, Zhengtao Liu, Martina Klevstig, Bani Mukhopadhyay, Mattias Bergentall, Resat Cinar, Marcus Ståhlman, Natasha Sikanic, Joshua K. Park, Sumit Deshmukh, Azadeh M. Harzandi, Tim Kuijpers, Morten Grøtli, Simon J. Elsässer, Brian D. Piening, Michael Snyder, Ulf Smith, Jens Nielsen, Fredrik Bäckhed, George Kunos, Mathias Uhlen, Jan Borén & Adil Mardinoglu

Corresponding authors: Jan Borén & Adil Mardinoglu, Chalmers University of Technology

Review timeline:

Submission date:	21 May 2017
Editorial Decision:	20 June 2017
Revision received:	04 July 2017
Editorial Decision:	17 July 2017
Revision received:	19 July 2017
Accepted:	24 July 2017

Editor: Maria Polychronidou

Transaction Report:

1st Editorial Decision

20 June 2017

The reviewers' recommendations are rather clear so I think that there is no need to repeat the points listed below. Of course, feel free to contact me in case you would like to discuss any particular point in further detail.

On a more editorial level, we would like to ask you to address the following issues:

- We have replaced Supplementary Information by the Expanded View (EV) format, in which a limited number of Supplementary Figures (typically ~5) are included in the article as EV figures. In this case all additional figures can be displayed as EV Figures. EV Figures should be provided as individual files and their legends should be included in the main text. For detailed instructions regarding expanded view please refer to our Author Guidelines.
- Datasets should be provided as individual .xls files labeled and cited as Dataset EV1, Dataset EV2 etc. The description of each dataset should be included in a separate tab in the corresponding .xls file.
- The link <http://sysmedicine.com/cns> providing access to the networks generated in the study is not working. For long term archival, we would also ask you to provide these files as EV Datasets or to upload them in a public repository. For more information please refer to our author guidelines for Data Deposition.

- The related unpublished study by Mardinoglu et al, should be cited in the text as "Mardinoglu et al, in preparation" (unless it has in the meantime been accepted, and is currently in press). For more information regarding citation of unpublished work you can refer to our Author Guidelines.

- Please provide a "standfirst text" summarizing the study in one or two sentences (approximately 250 characters), three to four "bullet points" highlighting the main findings and a "synopsis image" (211x157 pixels, jpeg format) to highlight the paper on our homepage.

- When you resubmit your manuscript, please download our CHECKLIST and include the completed form in your submission. *Please note* that the Author Checklist will be published alongside the paper as part of the transparent process.

REFEREE REPORTS

Reviewer #1:

The authors present an elegant study where co-expression networks (CNs) were constructed for 46 human tissues and liver cancer, followed up by detailed analysis of liver CNs and subsequently by validation studies in vitro and in vivo.

The manuscript is well written and the presentation is clear.

There are two issues the authors may consider elaborating further.

There is little information what samples were used to construct CNs. Specifically for liver which has been studied in more detail than other tissues in the study, one would expect that the type of samples included in construction of CNs would affect the co-expression patterns. How were the covariates such as gender, age, amount of liver fat been taken into account?

The method for construction of CNs is rather straightforward and based on Person correlation. No particular statistic was used to evaluate significance of associations and instead a cutoff value was used based on the ranking of associations. The motivation for this choice needs to be explained. Furthermore, Person correlation would lead to large number of spurious associations. Ideally, one would rather apply alternative methods for construction of CNs such as those based on partial correlation. The authors should explain better the reasoning for their choice and perhaps discuss the limitation in the Discussion.

Reviewer #2:

In this study, lee et al have generated tissue-specific co-expression networks for 46 major human tissues and human hepatocellular carcinoma (HCC), and deciphered their tissue-specific functions. The results indicate that hepatic fatty acid synthase (FASN) is co-expressed with a few liver-specific genes associated with de novo lipogenesis that usually occurs during non-alcoholic fatty liver disease (NAFLD) and HCC. In fact, their reduced expression was found to be associated with reduced FASN expression. In addition, the authors potentially identify the cannabinoid type-1 receptor (CB1R) as the upstream modulator of these genes. This work complements their previously documented findings that describe other genomic analysis methodologies (such as transcriptional regulatory and protein-to-protein interaction networks) to identify tissue-specific genes that are correlate with functionality during health and disease.

Overall, these are interesting findings that provide compelling evidence for using such strategy to identify novel genes that are dysregulated during diseases. The authors provide a wealth of novel data supporting the fact that FASN expression is associated with PKLR, PNPLA3, and PCSK9, inhibition of which might be used as a potential treatment for NAFLD and HCC. Having said that, there are numerous issues that need to be addressed to improve the quality of the manuscript:

1. To identify the functional relationship between FASN and the hepatic-specific genes identified during NAFLD progression, the authors oddly decided to use a two-week high sucrose diet (HSD) mouse model. Since NAFLD is a well-known chronic condition that affects the liver, it would be important to show that a similar pattern of expression exists following a long-term HSD feeding. Moreover, beside evaluating hepatic triglyceride content (Fig. 5A), the authors should fully characterize their model by providing histological evaluation of the liver, assessing NASH score, determining serum ALT and AST levels, etc.
2. In accordance with the reduced FASN and its associated targets in PCSK9-null mice, are they also resistant to the development of hepatic steatosis on regular chow and HSD feeding?
3. The authors mentioned that the attenuation of FASN and its related targets in CB1R-null mice treated with DEN is associated with decreased tumor growth. However, no data representing tumor size are provided. I understand that they used samples that have been already described (Mukhopadhyay et al., 2015), but at least correlations between tumor size and gene expression should be added.
4. Since peripheral CB1R blockade has been shown to ameliorate HCC (Mukhopadhyay et al., 2015), it would be important to determine whether this effect is associated with reduced FASN expression and its related genes.
5. Identifying PKLR, PCSK9, and PNPLA3 genes to be associated with FASN expression as well as NAFLD and HCC progression is a major novelty of the current paper. Indeed, the authors conclude that targeting these genes/proteins by using a chemical compounds or monoclonal antibodies could be considered as an effective treatment for such diseases. Therefore, to further validate their conclusion, they should consider testing PCSK9 inhibitor for reversing NAFLD or attenuating HCC in their mouse models.

Minor comments:

1. It isn't clear whether the data in all figures are presented as mean {plus minus} sem.
2. In figure 5D & E the gene expression is normalized to % of HCC, why? Isn't it better to describe it against the expression in the noncancerous tissues?

1st Revision - authors' response

04 July 2017

RESPONSE TO REVIEWER COMMENTS:

Reviewer #1:

The authors present an elegant study where co-expression networks (CNs) were constructed for 46 human tissues and liver cancer, followed up by detailed analysis of liver CNs and subsequently by validation studies in vitro and in vivo. The manuscript is well written and the presentation is clear. There are two issues the authors may consider elaborating further.

We would like to thank the reviewer for careful reading of our manuscript and providing constructive comments.

There is little information what samples were used to construct CNs. Specifically for liver which has been studied in more detail than other tissues in the study, one would expect that the type of samples included in construction of CNs would affect the co-expression patterns. How were the covariates such as gender, age, amount of liver fat been taken into account?

From GTEx database, we obtained RNA-seq data of tissues of all deceased donors and we used data as it is, considering that they are from normal population without selection bias. Therefore, co-expression networks based on these cohorts are expected to identify more general functional relationships rather than associations with a specific clinical indication.

Considering the comment of the reviewer, we discussed the limitation of cohorts used in this study in the Discussion section. We have not considered the effect of covariates while constructing CNs because their effects would be distinct by tissues, especially reproductive tissues, and may increase biases while comparing different tissue CNs.

The method for construction of CNs is rather straightforward and based on Person correlation. No particular statistic was used to evaluate significance of associations and instead a cutoff value was used based on the ranking of associations. The motivation for this choice needs to be explained.

Furthermore, Person correlation would lead to large number of spurious associations. Ideally, one would rather apply alternative methods for construction of CNs such as those based on partial correlation. The authors should explain better the reasoning for their choice and perhaps discuss the limitation in the Discussion.

Considering unbiased investigations across diverse tissues, we preferred to choose straightforward Pearson's correlations. Partial correlation would be alternative method, but it is still limited given that controlling random variables for co-expressions are not known and would increase biases by different tissues. We have used cut-off value, but it is strict that average cut-off correlation value was 0.576 and p-values of cut off correlations among all 46 tissues were less than $1E-5$.

We agree that spurious associations would be found from Pearson's correlations and it is discussed in the Discussion section. In order to avoid possible bias based on Pearson's correlation method, we have checked FASN-coexpressed target genes in HCC tumor, such as PKLR, PNPLA3, and PCSK9, by not only raw expression values but also log-transformed values, checked their expressions between patient groups stratified by FASN expressions (Figure 3B) and found significant correlation of genes with the expression of FASN.

Based on the suggestion of the reviewer, we included the following section to the paper.

"Correlation analysis is used to identify co-expressions between different genes based on mRNA expression data. Although co-expression does not necessarily indicate a relationship among transcript levels, functional relationships between encoded protein coding genes have been shown in our study. Co-expression analysis is applied to identify important alterations related to lipid synthesis, and promotion of cell survival and cell growth. In order to avoid possible false positives from co-expressed genes, we applied strict cut-off of Pearson's correlation coefficients, but it would be necessary to develop advanced method to identify co-expressed genes for smaller number of false positives. Here, we calculated correlations of gene expressions using normal tissue samples obtained from general population (i.e. GTEx) without selection bias, thus expected to identify general functional relationships, not only associated with a specific clinical indication. However, it is noteworthy that RNA-seq data generated for large and extensively-phenotyped cohorts would improve our understanding of functional relationships between genes in a specific clinical indication."

Reviewer #2:

In this study, Lee et al have generated tissue-specific co-expression networks for 46 major human tissues and human hepatocellular carcinoma (HCC), and deciphered their tissue-specific functions. The results indicate that hepatic fatty acid synthase (FASN) is co-expressed with a few liver-specific genes associated with de novo lipogenesis that usually occurs during non-alcoholic fatty liver disease (NAFLD) and HCC. In fact, their reduced expression was found to be associated with reduced FASN expression. In addition, the authors potentially identify the cannabinoid type-1 receptor (CB1R) as the upstream modulator of these genes. This work complements their previously documented findings that describe other genomic analysis methodologies (such as transcriptional regulatory and protein-to-protein interaction networks) to identify tissue-specific genes that are correlate with functionality during health and disease.

Overall, these are interesting findings that provide compelling evidence for using such strategy to identify novel genes that are dysregulated during diseases. The authors provide a wealth of novel data supporting the fact that FASN expression is associated with PKLR, PNPLA3, and PCSK9, inhibition of which might be used as a potential treatment for NAFLD and HCC. Having said that, there are numerous issues that need to be addressed to improve the quality of the manuscript:

We would like to thank the reviewer for careful reading of our manuscript and providing constructive comments.

1. To identify the functional relationship between FASN and the hepatic-specific genes identified during NAFLD progression, the authors oddly decided to use a two-week high sucrose diet (HSD) mouse model. Since NAFLD is a well-known chronic condition that affects the liver, it would be important to show that a similar pattern of expression exists following a long-term HSD feeding. Moreover, beside evaluating hepatic triglyceride content (Fig. 5A), the authors should fully characterize their model by providing histological evaluation of the liver, assessing NASH score, determining serum ALT and AST levels, etc.

In our study, we investigated the relationship between hepatic steatosis and the expression of FASN and our target genes. In order to show this association, we fed the mice with HSD for two weeks and measured the Hepatic TG levels as well as the expression of the genes. We also found a reference reporting that it is reasonable for generation of a NAFLD mouse model after 2-3 weeks of HSD (“Koteish A, Diehl A M. Animal models of steatosis, Seminars in liver disease (2001).”)

In another independent mouse experiment, we fed the mouse with HSD for longer period and measured the hepatic steatosis and the expression of the genes using RT-PCR. We found very similar results. These liver tissue samples will be sent out for RNA sequencing and the global metabolic alterations will be studied in a follow up study.

2. In accordance with the reduced FASN and its associated targets in PCSK9-null mice, are they also resistant to the development of hepatic steatosis on regular chow and HSD feeding?

It has been reported that lipid accumulation in hepatocytes of PCSK9 KO mice was markedly reduced under both chow and high-cholesterol diets, revealing that PCSK9 deficiency confers resistance to liver steatosis (PMID: 18666258).

Moreover, Ruscica et al. (Ruscica et al, 2016) have recently associated circulating PCSK9 levels with accumulated liver fat. In 201 consecutive patients biopsied for suspected NASH, liver damage has been quantified by NAFLD activity score, circulating PCSK9 by ELISA, and hepatic mRNA by qRT-PCR in 76 of the patients. Circulating PCSK9 has been found to be significantly associated with steatosis grade, necroinflammation, ballooning, and fibrosis stage. Circulating PCSK9 has also been found to be significantly associated with hepatic expression of SREBP-1c and FASN, whereas PCSK9 mRNA levels have been found to be significantly correlated with steatosis severity and hepatic APOB, SREBP-1c and FASN expression.

Data in the literature supporting our findings presented in our manuscript.

3. The authors mentioned that the attenuation of FASN and its related targets in CB1R-null mice treated with DEN is associated with decreased tumor growth. However, no data representing tumor

size are provided. I understand that they used samples that have been already described (Mukhopadhyay et al., 2015), but at least correlations between tumor size and gene expression should be added.

The relationship between FASN and tumor size in mice may not have any significant correlation as FASN is a switching function (on or off) with tumor. In consistent with our hypothesis we have found that FASN is upregulated in mice which had detectable tumor in liver of mice.

Moreover, we have tumor size values from individual mice, but the FASN gene expression was measured by RNAseq using pooled tissue.

4. Since peripheral CB1R blockade has been shown to ameliorate HCC (Mukhopadhyay et al., 2015), it would be important to determine whether this effect is associated with reduced FASN expression and its related genes.

We observed that in the absence of CB1R, there is very low level FASN mRNA in the control and tumor of CB1R^{-/-} mice. It further strengthens our hypothesis and FASN is detectable in higher level in HCC mice when tumor is present.

5. Identifying PKLR, PCSK9, and PNPLA3 genes to be associated with FASN expression as well as NAFLD and HCC progression is a major novelty of the current paper. Indeed, the authors conclude that targeting these genes/proteins by using a chemical compounds or monoclonal antibodies could be considered as an effective treatment for such diseases. Therefore, to further validate their conclusion, they should consider testing PCSK9 inhibitor for reversing NAFLD or attenuating HCC in their mouse models.

This is a very good point. However, there is no inhibitor available for any of the target genes including PKLR, PCSK9, and PNPLA3. We have started to work on the development of the small molecule and we will test it as soon as we have the first compounds available.

Minor comments:

1. It isn't clear whether the data in all figures are presented as mean {plus minus} sem.

Data are presented as the means ± standard errors

2. In figure 5D & E the gene expression is normalized to % of HCC, why? Isn't it better to describe it against the expression in the noncancerous tissues?

We wanted to show the link between expressions of the genes not only between wild type HCC tumor and noncancerous samples but also in CB₁R knockout HCC tumor and wild type HCC tumors.

Dear Adil,

Thank you for sending us your revised manuscript. We have now heard back from reviewer #2 who was asked to evaluate your study. As you will see below, the reviewer is satisfied with some of the modifications made but has some remaining concerns, which we would ask you to address in a revision of the manuscript.

While we think that the experiments in mice fed a long-term HSD (point #1 of the reviewer) are not mandatory for acceptance of the paper, we think that points #4 and #5 need to be addressed, since they would indeed enhance the conclusiveness of the study.

On a more editorial level, I would like to draw your attention to the following points:

- I have made some changes in the standfirst text, bullet points and abstract (see attached file). Could you please let me know whether you agree with these changes or if you would prefer to further edit the text?

- I would also suggest the slightly modified title: "Network analyses identify liver-specific targets for treating liver diseases".

Please resubmit your revised manuscript online, with a covering letter listing amendments and responses to each point raised by the referees. Please resubmit the paper ****within one month**** and ideally as soon as possible. If we do not receive the revised manuscript within this time period, the file might be closed and any subsequent resubmission would be treated as a new manuscript. Please use the Manuscript Number (above) in all correspondence.

REFeree REPORT

Reviewer #2:

1. The authors have decided not to provide data, which evaluate their HSD-induced hepatic steatosis mouse model. Instead, they have chosen to cite a paper describing the model. As I mentioned before, NAFLD is a chronic condition, and I still believe that validating their data in a new set of animals would greatly add to the impact of the current paper.
2. Thank you for clarifying this point.
3. I am satisfied with the explanation provided.
4. The authors did not determine the expression levels of FASN in mice treated with a peripheral CB1 receptor blocker, which has been shown to reduce HCC occurrence.
5. There are currently two PCSK9 inhibitors, Evolocumab (Repatha) and Alirocumab (Praluent) that have been developed and tested, so stating that there is no inhibitor available is basically not too accurate.

RESPONSE TO REVIEWER COMMENT:

We would like to thank the reviewer for careful reading of our manuscript and providing constructive comments.

1. The authors have decided not to provide data, which evaluate their HSD-induced hepatic steatosis mouse model. Instead, they have chosen to cite a paper describing the model. As I mentioned before, NAFLD is a chronic condition, and I still believe that validating their data in a new set of animals would greatly add to the impact of the current paper.

We performed additional independent mouse experiment, generated RNAseq data and confirmed our results. Data has been presented in the thesis of Mattias Bergentall and will be published in a follow up study. Please let me know if you want me to send the thesis or the draft of the paper. Considering that we had similar results, we decided not to include in this paper.

2. Thank you for clarifying this point.

3. I am satisfied with the explanation provided.

4. The authors did not determine the expression levels of FASN in mice treated with a peripheral CB1 receptor blocker, which has been shown to reduce HCC occurrence.

We have already included the expression of FASN in the noncancerous and tumor samples obtained from wild type and CB1 KO mice in Figure 5. Moreover, we had tumor size values from individual mice and the FASN gene expression was measured in these set of mice.

5. There are currently two PCSK9 inhibitors, Evolocumab (Repatha) and Alirocumab (Praluent) that have been developed and tested, so stating that there is no inhibitor available is basically not too accurate.

Thank you so much for pointing this. The drugs mentioned by the reviewer are not inhibitors. They are very well known monoclonal antibodies and target the circulation rather than liver tissue itself. That is the reason; there is a need for the development of inhibitors not only for PCSK9 but also for other tissues. We are developing PKL inhibitors and Pfizer is developing PCSK9 inhibitors. We are also filing a patent siRNA for PKL. Should be filed before August 15.

3rd Editorial Decision

24 July 2017

Thank you again for sending us your revised manuscript. We are now satisfied with the modifications made and I am pleased to inform you that your paper has been accepted for publication.

Corresponding Author Name: Adil Mardinoglu

Manuscript Number: MSB-17-7703